# Vitamin D and Omega-3 (Fatty Acid) Supplementation in Pregnancy for the Primary Prevention of Food Allergy in Children-Literature Review

**DOI:** 10.3390/children10030468

**Published:** 2023-02-27

**Authors:** Gavriela Feketea, Maria Kostara, Roxana Silvia Bumbacea, Emilia Vassilopoulou, Sophia Tsabouri

**Affiliations:** 1Department of Pharmacology, Toxicology and Clinical Pharmacology, University of Medicine and Pharmacy, 400337 Cluj-Napoca, Romania; 2Department of Pediatrics, “Karamandaneio” Children’s Hospital of Patra, 26331 Patras, Greece; 3Department of Pediatrics, University Hospital of Ioannina, 45500 Ioannina, Greece; 4Allergology Department, “Carol Davila” University of Medicine and Pharmacy, 050474 Bucharest, Romania; 5Department of Allergology and Clinical Immunology, “Carol Davila” Nephrology Clinical Hospital, 010731 Bucharest, Romania; 6Department of Nutritional Sciences and Dietetics, International Hellenic University, 57400 Thessaloniki, Greece; 7Child Health Department, University of Ioannina School of Medicine, 45500 Ioannina, Greece

**Keywords:** Vitamin D, omega-3 fatty acids, food allergy, children, prevention, pregnancy

## Abstract

During the last decades the prevalence of food allergy (FA), an adverse immune response to a specific food antigen, has risen, with negative effects on the quality of life (QoL) of many children and their families. The pathogenesis of FA is complex, involving both genetic and environmental factors. SPINK5, STAT6, HLA and FOXP3 are some of the genes that are reported to be implicated in FA development. Regarding environmental factors, particular interest has been focused on modification of the dietary habits of pregnant women for the primary prevention of FA. Specifically, Vitamin D and omega-3 (Ω-3) fatty acid supplementation during pregnancy may influence the development of FA in the offspring. Vitamin D is a hormone with various actions, including mediation of the immune system, reducing the production of inflammatory cytokines and promoting tolerance. Vitamin D deficiency in pregnancy suppresses T-regulatory cells in the fetus, and Vitamin D supplementation might protect against FA development. Dietary Ω-3 fatty acids are found mainly in fish and vegetable oils. They are beneficial for human health, playing a role in the immune system as anti-inflammatory agents, and providing cell membrane stabilization with inhibition of antigen presentation. It is documented that maternal supplementation with Ω-3 during pregnancy may protect from allergic sensitization in the children. The aim of this literature review was to explore the potential preventive role of maternal supplementation during pregnancy with Vitamin D and Ω-3 in the development of FA in the offspring. With the prevalence of FA rising, all the possible protective mechanisms and measures for FA prevention need to be explored, starting with those that can be modified.

## 1. Introduction

Food allergy (FA) is an adverse immune response to a specific food antigen, usually protein, which emerges reproducibly in children with intolerance [1,2]. This immunological disruption can be either IgE or non-IgE mediated. IgE-mediated reactions may present with symptoms affecting the skin, and the circulatory, respiratory and gastrointestinal (GI) systems. Νon-IgE-mediated and mixed IgE- and non-IgE-mediated reactions occur commonly with GI symptoms (vomiting, diarrhea, abdominal pain, bloody stools) [3]. Anaphylaxis, urticaria and angioedema are recognized to be secondary to an Ig-E mediated mechanism. Food protein-induced disorders, including food protein-induced allergic proctocolitis (FPIAP), food protein-induced enteropathy (FPIE) and food protein-induced enterocolitis syndrome (FPIES) are due to a non-IgE-mediated mechanism [4]. While any food can potentially trigger an allergic response, eight foods cause the majority of allergic reactions: milk, egg, fish, peanut, tree nuts, shellfish, wheat and soy [5]. In individuals with FA, sensitization to certain food allergens causes an inappropriate inflammatory immune response [6]. Regardless of the mechanism implicated, FA constitutes a major health concern, which affects the quality of life (QoL) of the children with FA, but also their families [7]. In the last decades, the prevalence of FA has risen, especially in the western world, and currently about 8% of children are reported to be affected [8,9]. To date, no effective treatment has been established, and strict allergen avoidance continues to be the main advice for management, which is difficult to apply [10]. The pathogenesis of FA is complex, and includes several genetic and environmental factors [11]. Family and twin studies have ascertained the genetic basis of FA [12], and several genes, including SPINK5, STAT6, HLA and FOXP3, have been implicated in its development [13,14,15]. Regarding environmental factors, these include pollutants, infections, exposure to sunlight, breastfeeding, and maternal diet and dietary supplements during pregnancy and lactation [16]. Recent studies have focused on maternal dietary habits during pregnancy in the prevention of FA [17], with special attention to the intake of supplements, such as Vitamin D and omega-3 (Ω-3) fatty acids.

Vitamin D is a fat-soluble hormone with pleiotropic effects which can have an impact on FA development in various ways. Firstly, it acts as an immune system modulator [18], as it restrains the Th1/Th2 responses [19] by reducing the production by T cells of inflammatory cytokines [20,21], which are responsible for the allergic response [22]. In addition, Vitamin D inhibits T cell proliferation and promotes the induction of T-regulatory cells which promote tolerance [23,24]. Vitamin D deficiency in pregnancy is responsible for T-regulatory cell suppression, and is a risk factor for FA development in the offspring [25]. Vitamin D supplementation in pregnant women was shown to be associated with a lower rate of detection of specific IgE to food antigens in their children at the age of five years [26]. Sufficiency of Vitamin D appears to exert a protective role in the development of atopy and allergic diseases [27,28]. The Ω-3 fatty acids are a family of biologically active unsaturated fatty acids. Long chain polyunsaturated fatty acids (LC-PUFAs) play a regulatory role in the immune system, as they affect cell signaling and antigen presentation, protecting against inflammation [29]. In addition, correlation has been demonstrated between high fish consumption in pregnancy and allergy development in the offspring [30]. The aim of this literature review was to explore the role of Vitamin D and Ω-3 supplementation during pregnancy in the development of FA in the offspring.

### Literature Review Strategy and Methods

A search was made of peer-reviewed literature published between 1959 and December 2022 in PubMed, Scholar Google, Cochrane and EMBASE, using combinations of the key words “food allergic reactions in children” and “Vitamin D”, “omega-3 fatty acids”, “food allergy prevention”, and “pregnancy”. The reference lists of the retrieved articles were checked for other relevant articles not found during the initial search. Personal collections of articles on the topic were also used to extend the search. The initial literature search identified 93 publications, including original studies, meta-analyses/randomized controlled trails (RCTs) and systematic reviews. Subsequently, the literature search and discussion were focused on, but not limited to, the results of double-blind placebo-controlled studies.

## 2. Vitamin D: Synthesis and Metabolism

Although vitamin D has been considered to be a micronutrient, it is a prohormone that, when transformed into its biologically active forms, regulates many physiological functions [31]. Figure 1 summarizes the metabolism of vitamin D in the body. The two major isoforms of vitamin D are vitamin D2, known as ergocalciferol, and vitamin D3, known as cholecalciferol. After exposure to UVB radiation in sunlight, vitamin D2 and D3 are synthesized in the skin from ergosterol and 7-dehydrocholesterol, respectively. Both isoforms are biologically inactive until they are delivered to the liver by vitamin D-binding protein (VDBP) and metabolized by vitamin D 25-hydroxylase (CYP2R1 and CYP27A1) to 25(OH)D (calcidiol), which is the major circulating form of vitamin D. The serum level of 25(OH)D is used as an indicator of vitamin D status in a human organism [31]. The 25(OH)D is further metabolized, mainly in the proximal tubule of the kidney by 25(OH)D 1α-hydroxylase (CYP27B1) to 1α, 25-dihydroxy vitamin D (1α,25[OH]_2_D, calcitriol), which is the recognized biologically active form of vitamin D [32]. Calcitriol then enters the circulation, where it binds to VDBP, and reaches target tissues, including intestine, bone and kidney, to regulate the absorption, mobilization and reabsorption, respectively, of calcium and phosphate [33].

Another metabolic route for vitamin D is by CYP11A1, which is a cytochrome P450 side-chain cleavage (P450scc) enzyme [34]. Vitamin D substitutes cholesterol as a substrate for CYP11A1, where more than 21 hydroxy-metabolites of vitamin D are produced [35].

## 3. Vitamin D: Biological Actions and Health Benefits

The main function of 1,25(OH)_2_D is regulation of calcium and bone metabolism [36], although its biological activities are broader, and include the regulation, proliferation and differentiation of a variety of cells, including keratinocytes, endothelial cells, osteoblasts, and lymphocytes [37]. Most of these biological functions are mediated via the vitamin D nuclear receptor (VDR) which acts as a transcription factor, regulating the transcription of target genes [38]. The CYP11A1 metabolites have antiproliferative, differentiating and anti-inflammatory actions in skin cells, comparable to that of calcitriol [39], they are involved in defense pathways against UVB-induced damage and oxidative stress [40] and they induce cell-specific anticancer effects [41].

## 4. Vitamin D and Immune Function

The effects of Vitamin D in immune system function are multifaceted, as shown in Figure 2A. It impedes B-cell proliferation and differentiation, and immunoglobulin secretion [42]. It inhibits T-cell proliferation and promotes the induction of T-regulatory cells [43,44]. Consequently, inflammatory cytokines, such as interleukin 17 (IL17) and IL21, are decreased, while anti-inflammatory cytokines, such as IL 10, are increased [45,46]. Monocyte production of the inflammatory cytokines, including IL 1, IL 6, IL 8, IL 12, and tumor necrosis factor-α (TNFα), is inhibited in the presence of vitamin D [47], and it also inhibits the differentiation and maturation of the dendritic cells [48].

## 5. Vitamin D: Sources

The most abundant source of vitamin D is sunlight; when skin is exposed to sunlight the 7-dehydrocholesterol absorbs UVB radiation and is converted to provitamin D3, which, as described above, is then converted to vitamin D3 [49]. Other sources of vitamin D3 (cholecalciferol) are animal foods and vitamin D3 supplements, while vitamin D2 (ergocalciferol) is derived from the intake of vegetable foods and vitamin D2 supplementation [50]. As exposure to the sun varies according to the season, time of day, latitude, altitude, skin pigmentation and other conditions, food sources are required to cover the required 15 μg daily intake set by the European Food Safety Authority (EFSA) [51,52]. Good sources of dietary vitamin D2 are oily fish and fish liver oils, mushrooms, reindeer lichen (*Cladonia rangiferina*), beef liver, eggs, dark chocolate and cheese. A variety of fortified foods are available, including dairy products, juices and breakfast grains, which aim to cover the recommended daily requirements for vitamin D [53].

## 6. Vitamin D Supplementation in Pregnancy; Nutritional Benefits and/or Prevention of FA

The recommended intake (RI) of vitamin D for pregnant women is similar to that for non-pregnant women, namely 600 IU/day [54,55]. During the first year of life, the RI of dietary vitamin D is 400 IU/day, and between 1 and 18 years, 600 IU/day [56]. Pregnant women are advised to consume foods with high nutrient density, to ensure adequate levels of vitamin D, but in practice, the estimated daily vitamin D intake during pregnancy may be higher or lower than the current recommendations, depending on the population under study [57,58,59]. Vitamin D supplementation alone in pregnancy, as part of routine antenatal care, is not recommended [60].

Several studies have documented an association of adequate vitamin D intake during pregnancy with a decreased risk of wheezing in the offspring [61,62,63,64], and others have investigated the association between vitamin D consumption during pregnancy and development of asthma and allergic rhinitis in the offspring [65,66,67,68,69,70,71]. A systematic review published by the European Academy of Allergy and Clinical Immunology (EAACI) demonstrated a reduction in asthma in the offspring following maternal vitamin D supplementation in pregnancy [72].

Vassallo and colleagues showed that birth in the fall or winter, when vitamin D levels are lowest, was associated with a higher risk of presenting at the emergency department with food-related acute allergic symptoms [73]. Based on the link with season of birth and latitude, epidemiological evidence suggests low UVB exposure as a risk factor for FA, but the relationship vitamin D and FA is unclear, and is certainly nonlinear [74].

To date, only a few studies assessed the effectiveness of vitamin D intake during pregnancy in preventing the development of FA in the offspring, and their results were inconsistent. Evidence on the potential protective role of vitamin D in the development of FA is derived from studies assessing vitamin D dietary intake and vitamin D supplementation during pregnancy, from studies measuring maternal vitamin D status or cord blood vitamin D level, and from interventional studies.

Regarding evaluation of the maternal diet for the prevention of allergic disease in general, studies have been made of individual nutrients, including vitamin D, and of particular foods rich in vitamin D and/or dietary patterns, such as the Mediterranean diet (MedDiet). One recent retrospective, observational, multicenter, case-control study showed that vitamin D supplementation appeared to reduce the risk of FPIAP, in the context of satisfactory adherence to the MedDiet [75]. The question is whether the observed associations are due to the foods and/or vitamin D supplement, or whether the effect is part of an overall nutritional composition of the maternal diet, a “food synergy” [76]. Other studies also suggest the possible effectiveness of a MedDiet during pregnancy for the prevention of atopy, sensitization and/or FA [77,78]. In an attempt to develop an index of maternal diet during pregnancy, associated with reduced odds of “any allergy excluding wheeze”, Venter and colleagues showed no association of any maternal dietary index with FA [17]. A Japanese prospective cohort study that included approximately 100,000 pregnant women detected no clear association between vitamin D intake during pregnancy and the development of FA in the offspring at the age of 1 year [58]. The relationship between vitamin D supplementation and/or Vitamin D inadequacy and the severity of different types of FA has not been widely explored. In one recent study, the consumption of multivitamins during pregnancy was shown to be correlated with longer duration of symptoms in children with FPIAP [79].

A EAACI systematic review reported that there is currently no evidence to support vitamin D supplementation trials in pregnancy for the prevention of FA development [72]. The US Centers for Disease Control (CDC) guidelines for pregnancy do not include recommendation for vitamin D supplement [80].

The results from observational studies on infant FA and maternal levels of vitamin D during pregnancy or from cord blood vitamin D levels are contradictory. Chiu and colleagues, studying children aged 0 through 4 years from a Taiwanese birth cohort (PATCH), reported an inverse association between cord blood 25(OH)D levels and milk sensitization at the age of 2 years [81]. Moreover, a deficient 25(OH)D level (<20 ng/mL) was significantly associated with a higher prevalence of sensitization to the three most common food allergens (milk, egg and wheat) at age 1.5 and 2 years in comparison with >20 ng.mL 25(OH)D levels [82]. Nwaru and colleagues proved, in a Finnish cohort, that increasing intake of Vitamin D during pregnancy was inversely associated with sensitization to milk, egg, wheat and fish at the age of 5 years [26]. A prospective cohort study in Chinese infants, suggested that cord blood 25(OH)D3 at an insufficient level (20 ng/mL ≤ 25(OH)D3 < 30 ng/mL) is more closely associated with predisposition to FA in infants at 6 months than a deficient state (25(OH)D3 < 20 ng/mL) [83]. Cord blood 25(OH)D3 deficiency appears to have a protective effect on the incidence of FA at this age, while a sufficient 25(OH)D3 level in infant cord blood was an independent risk factor for FA [84]. The German LINA cohort study confirmed that higher maternal and cord blood levels of 25(OH)D3 are associated with a risk of FA or sensitization to food allergens in children in the first 2 years of life [85].

Only one RCT was identified that investigated the effect of vitamin D supplementation during pregnancy on the development of FA in the offspring (Table 1). Specifically, at 27 weeks’ gestation, 180 pregnant women received either no vitamin D, 800 IU ergocalciferol (D2) daily until delivery, or a single oral bolus of 200,000 IU cholecalciferol (D3) [61]. No significant difference between dosages was reported in the incidence of food-specific IgE or “doctor-diagnosed FA” in the offspring [61].

One RCT on vitamin D maternal intake (800 UI/day) during the breastfeeding period reported an association with an increased incidence of FA in the offspring [86]. Similarly, a RCT of daily vitamin D supplementation of either 400 IU or 12,000 IU from the age of 2 weeks, found that milk allergy occurred more often in infants randomized to higher vitamin D supplementation [87]. An increased risk has been observed of allergic sensitization in infants with high cord blood vitamin D status [87].

It has been suggested that the relationship between vitamin D levels and susceptibility to allergic inflammation shows a U-shaped curve, with both very high and very low levels increasing the risk of atopy/allergy development [88,89]. A prospective, population-based birth cohort within the Finnish Type 1 Diabetes Prediction and Prevention Study found that vitamin D supplement use was associated with an increased risk of cow’s milk allergy (CMA) in the offspring, whereas vitamin D intake from foods consumed during pregnancy was associated with a decreased risk of CMA [90]. Similarly, a Japanese prebirth cohort study suggested that a higher maternal intake of vitamin D during pregnancy may increase the risk of infantile eczema, but not of wheeze [91]. Information on dietary supplements that included vitamin D was not recorded, and this study did not take into consideration sunlight exposure status, and data on serum concentration of 25-OH D were not available.

**Table 1 children-10-00468-t001:** Randomized controlled trials of maternal supplementation with vitamin D or omega-3 polyunsaturated fatty acids (Ω-3 LC-PUFAs) during pregnancy and food allergy/sensitization to food allergens in the offspring.

Authors, Country	Maternal Characteristics	Intervention (Nutrient, Concentration, etc.)	Period of Intervention	Follow-up Age	Food Allergy (FA) Outcomes in the Offspring	Other Allergy Outcomes in the Offspring
**Vitamin D**						
Goldring et al., 2013 UK [61]	180 pregnant women, at 27 weeks’ gestation,	Either no vitamin D, 800 IU ergocalciferol (D2) daily until delivery, or a single oral bolus of 200,000 IU cholecalciferol (D3)	April to November 2007.	3 years	No significant difference reported in the offspring in food-specific IgE or “doctor-diagnosed FA”	No significant difference between groups of infants in ‘wheeze ever’, prevalence of eczema or atopy, baseline respiratory resistance, total IgE level, eNO or eosinophil count
**Omega-3**						
Dunstan et al., 2003 Australia [92]	98 pregnant women, atopic, (i.e., offspring at high risk of allergic disease) nonsmoking, at 20 weeks’ gestation,	Either four 1-g fish oil capsules per day, comprising a total of 3.7 g of Ω-3 PUFAs, with 56.0% as docosahexaenoic acid (DHA) and 27.7% as eicosapentaenoic acid (EPA), or four 1-g capsules of olive oil per day, containing 66.6% n-9 oleic acid and < 1% Ω-3 PUFAs, as a placebo	1999–2001	12 months	Infants in the fish oil group were three times less likely to be sensitized to egg allergen	Infants in the fish oil group were less likely to develop recurrent wheeze, persistent cough, diagnosed asthma, FA, angioedema, or anaphylaxis, but the differences were not statistically significant.
Furuhjelm et al., 2009 Sweden [93]	145 pregnant women at 25 weeks’ gestation, with at least one family member having allergic symptoms (i.e., offspring at high risk of allergic disease)	Nine capsules a day containing Ω-3 PUFAs (35% EPA, 1.6 g/day and 25% DHA, 1.1 g/day), or soya bean oil (58% linoleic acid (LA), 2.5 g/day and 6% a-linolenic acid (LNA), 0.28 g/day) as a placebo	2003–2005	12 months	FA significantly less frequent, and the risk of developing allergic sensitization to egg lower in the Ω-3 group	A lower period prevalence of IgE-associated eczema in the Ω-3 PUFAs group
Furuhjelm et al., 2011 Sweden [94]	145 pregnant women at 25 weeks’ gestation, with at least one family member having allergic symptoms (i.e., offspring at high risk of allergic disease)	Nine capsules a day containing Ω-3 PUFAs (35% EPA, 1.6 g/day and 25% DHA, 1.1 g/day), or soya bean oil (58% linoleic acid (LA), 2.5 g/day and 6% a-linolenic acid (LNA), 0.28 g/day) as a placebo	2003–2005,	24 months	IgE-mediated food reactions, significantly less frequent and positive skin prick tests (SPTs) to food lower in the Ω-3 group	No difference between groups for “any asthma,” IgE-associated asthma, “any eczema,” “any rhino-conjunctivitis,” IgE-associated rhino-conjunctivitis Significant association between higher proportions of Ω-3 PUFAs in maternal and infant phospholipids and lower frequency and less severity of allergic diseases
Palmer et al., 2012 Australia [95]	706 pregnant women at 21 weeks’ gestation	Three 500 mg capsules of fish oil concentrate daily, providing 800 mg of DHA and 100 mg of EPA, or three 500 mg vegetable oil capsules without Ω-3 PUFAs as a placebo.	2005–2007	12 months	No significant difference in IgE-mediated FA between groups. The incidence of sensitization to egg was lower in the Ω-3 PUFA group.	The incidence of IgE-associated eczema was lower in the intervention group, although not to a significant degree
Palmer et al., 2013 Australia [96]	706 pregnant women at 21 weeks’ gestation	Three 500 mg capsules of fish oil concentrate daily, providing 800 mg of DHA and 100 mg of EPA, or three 500 mg vegetable oil capsules without Ω-3 PUFAs as placebo	2005–2007	3 years	No significant difference between groups in IgE-mediated. No difference between groups in sensitization to at least one allergen, including egg.	A lower, but not statistically significant, incidence of eczema with sensitization in the Ω-3 PUFAs group No significant reduction in IgE-associated allergic disease.
Best et al., 2018 Australia [97]	706 pregnant women at 21 weeks’,	Three 500 mg capsules of fish oil concentrate daily, providing 800 mg of DHA and 100 mg of EPA, or three 500 mg vegetable oil capsules without Ω-3 PUFAs as placebo	2005–2007	6 years	No significant difference between groups in the risk of sensitization to egg, peanut, cashew	No difference between groups in the risk of ‘any’ IgE mediated allergic disease or ‘individual’ IgE mediated allergic disease symptoms (eczema, rhinitis, rhino-conjunctivitis or wheeze)

Liu and colleagues examined the association of the 25(OH)D concentration in cord blood on the development of food sensitization, defined as specific IgE > 0.35 kUA/L to common food allergens: milk, soy, egg, peanut, walnut, fish, shrimp, and wheat. Because of the small number of cases of FA diagnosed by a doctor (31/460), this study was limited to food sensitization, and no association was found between low cord blood level of vitamin D and detectable IgE to any food allergen by age 3 years [98].

## 7. Omega3 Fatty Acids: Synthesis and Metabolism

Ω-3 fatty acids are a family of biologically active unsaturated fatty acids (UFAs) with the first site of a carbon-carbon double bond close to the methyl terminus of the acyl chain in their chemical structure. Often Ω-3 fatty acids are described by a nomenclature based on the number of carbon atoms in the acyl chain, the number of double bonds, and the number of the position of the first double bond relative to the methyl carbon [99].

Four major Ω-3 fatty acids have so far been shown to be involved in the health status and in disease prevention, namely: a-linoleic acid (ALA), eicosapentanoic acid (EPA), docosapentanoic acid (DPA) and docosaexanoic acid (DHA). ALA is the simplest Ω-3 fatty acid [18:3(n-3)], and it is synthesized from linoleic acid [18:2(n-6)] by catalytic desaturation by Δ15-desaturase. Plants possess Δ15-desaturase enzyme, and can therefore synthesize ALA. Animal and humans, however, do not possess Δ15-desaturase, and both ALA and linoleic acid are considered as essential fatty acids. Although animals cannot synthesise ALA, they can metabolize it into the longer chain Ω-3 fatty acids, EPA [20:5(Ω-3)], DPA [20:5(Ω-3)] and DHA [22:6(Ω-3)] [100]. Τhese gradual metabolic reactions take place mainly in the liver, and involve desaturation and elongation by desaturases (Δ6 and Δ5), and elongases, and β-oxidases only during the conversion of DPA to DHA [101]. Conversion of ALA to EPA, DPA and DHA is poor due to the limited availability of desaturases [102,103]. The conversion of linoleic acid to arachidonic acid [20:4(n-6)] competes with the conversion of ALA to EPA, as the same enzymes are used. EPA, DPA and DHA are known as very long-chain Ω-3 PUFAs.

## 8. Omega3 Fatty Acids: Biological Actions-Health Benefits

A wide range of health benefits has been demonstrated from the consumption of very long-chain Ω-3 PUFAs. Adequate Ω-3 intake during pregnancy and infancy safeguards the membrane phospholipids and biochemical development in the brain and retina in infancy, ensuring vision maturation and cognition [104,105].

The brain effect of Ω-3 is life-long, as neuropsychiatric disorders, such as Parkinson’s disease [105] and mild cognitive impairment [106], have been reversibly linked to Ω-3 deficiency.

Dietary Ω-3 fatty acids act therapeutically on, and protectively against, several cardiovascular abnormalities, including arrhythmias, hyperlipidemias, atherosclerosis and thrombosis [107]. Their benefit to the cardiovascular system is attributed to the metabolic regulation of lipids and lipoproteins, anti-inflammatory effects, platelet function, arterial cholesterol delivery, vascular function and regulation of blood pressure [108].

## 9. Omega3 Fatty Acids and Immune Function

Ω-3 fatty acids have immune-modulatory functions. Their metabolites, prostaglandins, thromboxanes, protectins, reolvins and maresins, known as pro-resolving mediators, have strong immunoregulatory effects. Their production from Ω-3 fatty acids is orchestrated by cyclooxygenase, lipoxygenase and cytochrome P450 enzymes [109]. These enzymes also catalyze the -6 metabolism [110], and therefore Ω-3 competes with Ω-6 for their use. Overall, Ω3 fatty acids play an important role in modulating inflammation and allergic phenomenon (Figure 2B).

Ω-3 shows activity in a broad range of immune cells. In macrophages, which have a fundamental role as part of the innate immune system, Ω-3 provokes major alterations in macrophage gene regulation [111], affecting the production and secretion of cytokines and chemokines, the capacity of phagocytosis and polarization into activated macrophages [111]. Ω-3 and metabolites of Ω-3 modulate the function of neutrophils, increasing their migration, phagocytic capacity and production of reactive oxygen species (ROS) and cytokines [109,112].

Several studies indicate that the balance between Ω-3 and Ω-6 fatty acids limits the differentiation of CD4+ T-helper cells into Th17cells, thus improving symptoms in children with asthma [113], and reducing the severity of autoimmune disease [114]. Dietary Ω-3 fatty acids have been implemented successfully in the CD4+ T cell differentiation into Tregs, with reduction of the severity of allergic diseases such as atopic dermatitis (AD) [115].

## 10. Omega3 Fatty Acids: Dietary Sources

Although green leaves are poor in fat, approximately 50% of their fatty acids are in the form of ALA. Seeds such as linseed, chia and flaxseed contain 45–55% of ALA, while walnuts, soybean and rapeseed oil contain approximately 10% of ALA [116,117].

Optimum sources of the very long-chain Ω-3 PUFAs (EPA, DPA and DHA) are oily fish, such as mackerel, tuna, salmon and sardines, which store lipids in their flesh and provide approximately 1.5–3.5 gr of these fatty acids. Lean fish such as cod, store lipids in their liver and provide 0.2–0.3 gr of very long-chain Ω-3 PUFA [118].

Fish oil is derived from lean fish liver or oily fish flesh and consists of 30% of EPA and DHA. A fish oil capsule of one gram provides approximately 0.3 g of EPA plus DHA, but the relevant proportions of the very long-chain Ω-3 PUFA in a capsule might vary depending on the source; a capsule of oil originating from tuna fish is richer in DHA than in EPA, while one from cod liver is richer in DHA than EPA [119,120].

## 11. Omega3 Fatty Acid Supplementation during Pregnancy, and Food Allergy Prevention

During the last decades, the dietary habits in the westernized world have changed, resulting in the consumption of more Ω-6 than Ω-3 PUFAs. According to the EFSA, during pregnancy, to the current recommendations of 250 mg/day DHA and EPA for adults, should be added 100–200 mg/day DHA [121]. The general requirements for children have not yet been adequately established, but the assumption is that children also benefit from lower saturated fat, and higher PUFA intake [122]. A systematic review comparing the real-life intake of PUFAs with EFSA recommendations revealed that the intake is largely suboptimal in specific population groups, including pregnant women [123].

Ω-6 PUFAs are considered proinflammatory, as they contribute to the appearance of allergic symptoms, whereas Ω-3 PUFAs exert a protective effect in allergy development [124,125]. Animal studies have shown that nutrition with a high proportion of Ω-3 PUFAs, such as DHA and EPA, can reduce allergic symptoms in mice suffering from FA. Also in mice, a diet rich in DHA appears to play a protective role against allergic sensitization [126]. These findings are further supported by clinical studies which document that the consumption of Ω-3 PUFAs early in life may have an impact on the development of the immune system and the function of its cells, decreasing the inflammatory response [127]. Ω-3 fatty acids are therefore considered to be anti-inflammatory, as they have improved allergic symptoms and reduced inflammation [128]. In two studies, pregnant women who had allergies, and who received daily Ω-3 supplementation from the 25th week of gestation, gave birth to infants with a lower risk of FA during the first year, and the first 2 years of life, respectively, than those of pregnant women with allergy who did not receive Ω-3 supplements [93,94]. In another study, pregnant women at high risk for allergy, received Ω-3 supplementation from the 21th week of gestation; at the age of 12 months, fewer of their infants were sensitized to egg, compared to a control group [95], but there was no difference in FA between the two groups of infants, either at the age of 12 months [95] or in the first 3 years of life [96]. This finding was not supported by a comparative study in Australia, where pregnant women with a history of atopy received fish oil containing Ω-3 fatty acids in a high proportion, from the 20th week of gestation. The infants of mothers who received fish oil supplementation were less likely to have positive skin prick tests to egg at the age of 12 months than those in the control group [92]. An Icelandic study which compared children who had received fish oil supplementation from the age of six months or later, on a regular basis, with children who had not, suggested that the children taking fish oil presented a lower risk for FA. Fish oil supplementation during pregnancy, however, did not have a protective effect on FA development in the offspring [129]. In contrast, in families with a history of atopy, no differences were reported in allergy sensitization or the appearance of IgE mediated allergic symptoms up to the age of six years, between children whose mothers consumed 900 mg of Ω-3 fatty acids during pregnancy and those who did not [97]. One systematic review of the possible effect of Ω-3 consumption during pregnancy on the development of allergy in the offspring suggested that this consumption was beneficial [130]. In addition, a significant reduction in the first 12 months of life was reported in the incidence of “any positive SPT” “sensitization to egg,” and “sensitization to any food” [130]. Another systematic review on the same subject found no lower risk for any kind of allergy in infants whose mothers consumed Ω-3 supplementation during pregnancy, but the risk for allergic outcomes was lower for children whose mothers consumed fish in infancy [131]. In pregnant women with high adherence to MedDiet, high fish intake reduced the risk of FPIAP, while Ω3 LC-PUFA supplementation appeared to be a risk factor [75]. Conversely, in a lower socioeconomic population, Ω3 LC-PUFA supplementation has been proposed for allergy prevention [132], suggesting an association with the dietary pattern. The characteristics and results of RTCs focused on the relationship between Vitamin D/Ω-3 and FA are presented in Table 1.

The results of the above studies regarding the possible protective role of the consumption of Ω-3 fatty acids during pregnancy against the development of FA in the offspring are inconsistent. Further studies, with rigid methodology, are needed to elucidate this complex association.

## 12. Future Considerations

As Vitamin D and Ω-3 fatty acids both play multifaceted roles in the immune system function, further studies are needed to clarify whether their supplementation during pregnancy has a protective role in the development of allergies in the offspring, and in particular FA. Certain parameters need to be reconsidered, such as the predisposition to allergy, with clear definition of the high-risk and low-risk study populations, the sunlight exposure status, the overall nutritional composition of the maternal diet and food synergy. Subsequently, the examination would be interesting of the simultaneous intake of Vitamin D and Ω-3 fatty acid supplements in pregnancy, to investigate possible modification of the final outcome of their protective role in infant FA.

## 13. Conclusions

There is no current evidence to support vitamin D supplementation in pregnancy for primary prevention against FA in infants. Supplementation might be protective in mothers with identified deficient intake or low serum levels. The relationship between Vitamin D and FA appears to be U-shaped, and therefore special attention is needed in the administration of Vitamin D in pregnancy for preventive reasons. Despite inconsistent evidence, it appears that fish oil supplementation during pregnancy could exert a protective influence against the development of FA. The appropriate population, the potential contribution of different dietary patterns and the relationship with other nutrients (i.e., food synergy), need to be further investigated. Dietary interventions, although not many, seem to have more firm results when adequate levels of fish or vitamins D is received under a healthy MedDiet, underlying the significance of the synergies among nutrients. There is a need to review real-life practices during pregnancy regarding the intake of vitamin D and Ω-3 PUFAs.

## Figures and Tables

**Figure 1 children-10-00468-f001:**
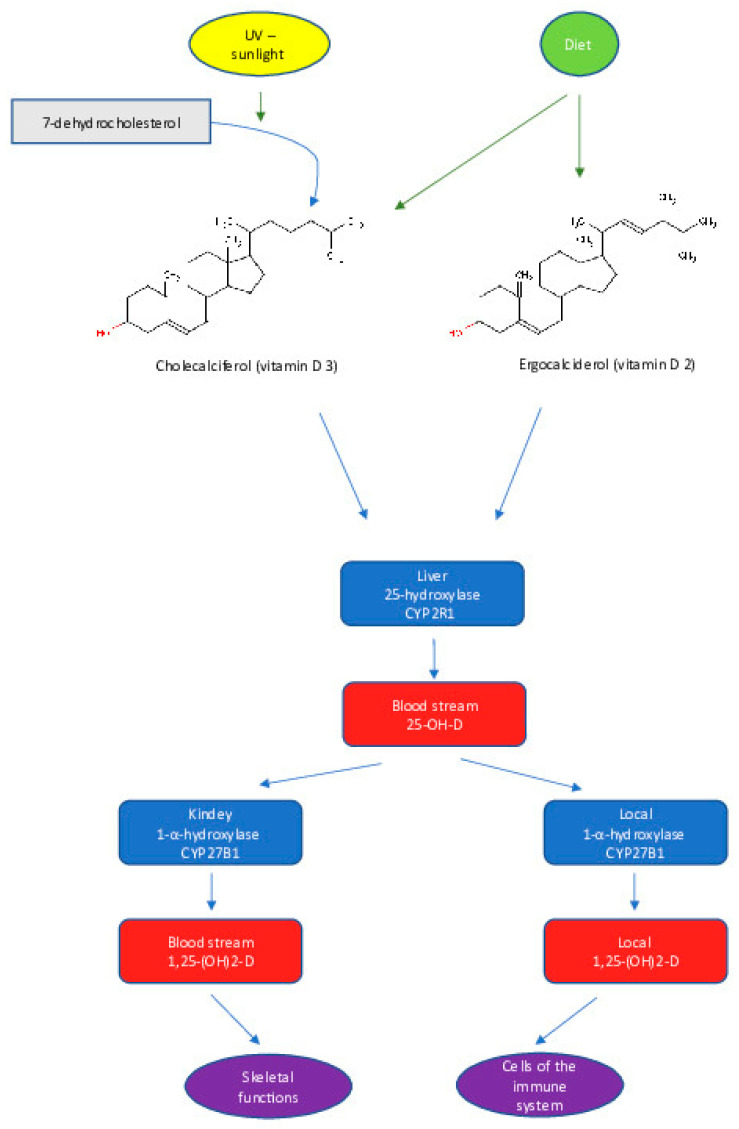
Vitamin D—sources and metabolism.

**Figure 2 children-10-00468-f002:**
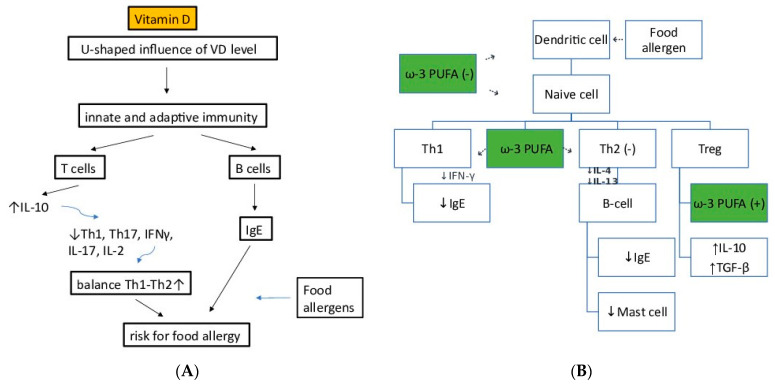
(**A**) The effect of polyunsaturated fatty acids (PUFAs) on food allergy. Vitamin D status affects the inhibitory or stimulatory response of the T cells, and B cells. The allergic reactions are affected through modulation of immune mediators such as IgE and pro- and anti-inflammatory cytokines (interleukins [IL-2, IL-10, IL-17]. (**B**) The effect of omega-3 (Ω-3) poly unsaturated fatty acids (PUFAs) on food allergy. The colour of the arrows and text indicate evidence obtained from clinical, in vivo or in vitro (green) data. The + or—indicates whether the observed effect is an inhibitory or stimulatory response of a certain cell type. Note that clinical and in vivo arrows indicate the observed end stage effects only; this may not be a reflection of the direct effect of PUFAs on the target cells. Therefore, the components could actually target a cell group earlier in the pathway. ⇢⇢↓IL-4 ↓IL-23.

## Data Availability

No new data were created.

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
