# Peer review of "Vitamin D and Omega-3 (Fatty Acid) Supplementation in Pregnancy for the Primary Prevention of Food Allergy in Children-Literature Review"

_children, 2023, doi:10.3390/children10030468_

Round 1

Reviewer 1 Report

While the authors offered a very general view of the possible cause and effect relationship between vitamin D/omega-3 and food allergies, there was very little mention (if any) on:

·         the types and severity of food allergies, which should at least be addressed in this paper, even if not much has been explored.

·         For example, are there certain types of food allergies that are more prone to be developed due to the lack of omega-3 intake? What was the severity of food allergies associated with the deficiencies of these 2 nutrients?

The manuscript was also lacking sufficient content suitable for a review article. There were also no figures or tables that would help readers navigate this review, which should be a strong consideration given the length of this paper.

The authors should revise their manuscript and add the appropriate content (including tables/figure summary) before submitting this as a review article.

Author Response

Comments and Suggestions for Authors from the Reviewer 1

While the authors offered a very general view of the possible cause and effect relationship between vitamin D/omega-3 and food allergies, there was very little mention (if any) on:

  • the types and severity of food allergies, which should at least be addressed in this paper, even if not much has been explored.
  • For example, are there certain types of food allergies that are more prone to be developed due to the lack of omega-3 intake? What was the severity of food allergies associated with the deficiencies of these 2 nutrients?

Thank you for your comment. In the new version of the manuscript, we have added examples of relationship between VD and type / severity of food allergies.

Additionally, in Figure 2 (A and B) we summarize the mechanisms by which vitamin D /ω-3 probably have immunomodulatory effects that alleviate the food allergic reaction in the various steps.

The manuscript was also lacking sufficient content suitable for a review article. There were also no figures or tables that would help readers navigate this review, which should be a strong consideration given the length of this paper.

Thank you for your suggestion. The authors included a Table and 2 Figures in the manuscript to make the article more easily readable.

The authors should revise their manuscript and add the appropriate content (including tables/figure summary) before submitting this as a review article.

We thank you for the useful suggestions and constructive report aiming at improving the scientific quality of our article. We agree with the points raised and in the new revised version of our paper we have implemented modifications addressing these points. A professional medical writer has revised the document to ensure sufficient English quality.

Comments and Suggestions for Authors from the Reviewer 2

In line 17 the pathogenesis of, please mention genetic factors here just like environmental factors.

Thank you for your comment. Genetic factors are now mentioned in the text.

How vitamin D deficiency affect T- cells and what will be its effect on child and mother health.

Thank you for your comment. Vitamin D deficiency in pregnancy is responsible for T-regulatory cell suppression and is a risk factor for FA development in the offspring. Vitamin D supplementation in pregnant women was associated with a lower risk of detection of specific IgE to food antigens in their children at the age of five years.

Add a conclusive line in the end of abstract summarizing the whole impact of your article.

Thank you for your comment a conclusive line has been added at the end of the abstract.

In line 43 and 44 “Regarding the environment, recent studies have indicated an effect of modification of maternal exposures during pregnancy, particularly dietary habits, in the prevention of FA” please rephrase it and how dietary habits come under the environmental studies.

Thank you for your comment. The sentence has been rephrased.

Please recheck and set the sentence in line 45 “Of special interest, are nutritional interventions, such as Vitamin D and omega-3 (Ω-3) supplementation during the prenatal periods”.

Thank you for your comment. The sentence has been rechecked.

Add types of food allergies and their prevalence national and international level as well as add a graph representing the food allergy in different region or countries.

Thank you for your comment. The exact prevalence of FA is difficult to determine because the characteristics of FA differ among races, ages and regions; FA is associated with geographical and dietary differences and countless other unknown factors. Unfortunately, well established data on the prevalence of food allergy in various regions is lacking, while there is a wide range among clinically diagnosed food allergies and self-reported food allergies. Therefore we cannot develop a diagram with well determined data on the prevalence of food allergy, as these are not provided by the literature as it happens with other allergic disease (for instance asthma) https://doi.org/10.1186/s12887-020-02115-8, DOI:10.1007/s11882-020-0898-7

Add some practical examples of food allergens and their types if possible.

Thank you for your suggestion.  In the new version of the manuscript, we have added examples of food allergens and more frequent food allergies in children.

Rewrite this paragraph “Several studies have documented that adequate vitamin D intake during pregnancy was associated with a decreased risk of wheezing in the offspring [51-56], and other studies have investigated the association between vitamin D consumption during pregnancy and development of asthma and allergic rhinitis in the offspring [57-63]. A European Academy of Allergy and Clinical Immunology (EAACI) systematic review noted a reduction in asthma in the offspring following maternal vitamin D supplementation in pregnancy”.

Thank you for your comment. The sentence has been rephrased.

Remove irrelevant material only focus on your allergens and vitamin D impact.

Thank you for your suggestion. We focused on food allergy and removed articles referring to other disorders.

Explain how food allergen effects immune system?

Thank you for your suggestion. In the new version of the manuscript, we have added “In individuals with food allergy, sensitization to certain food allergens causes in inappropriate inflammatory immune responses.”

Draw flow chart for vitamin D activation and also add thesis process site/ organ name.

Thank you for your comment. We have drawn a flow chart for vitamin D activation (figure 1 ).

Please avoid to use abbreviations and write full forms.

Thank you for your suggestion. In the new version of the manuscript, we have avoided some of the abbreviations and have kept them for the laboratory data.

Add recommended dose of vitamin D for children per day.

Thank you for your suggestion. In the new version of the manuscript, we have added the recommended dose of vitamin D for children per day.

Add chemical structure for all type of omega-3 fatty acid and draw flow chart for its mechanism.

Thank you for your comment. We have drawn a flow chart for omega-3 fatty acid mechanism (Figure 2).

Add full form of omega-3 fatty acid especially in headings.

Thank you for your suggestion. In the new version of the manuscript, we have used full form of omega-3 fatty acid in the headings.

Add recommended dose of omega-3 fatty acid for adults and children.

Thank you for your comment. In the new version of the manuscript, we have added: “The general requirements for children have not yet been adequately established. However, the assumption is that children also benefit from a lower saturated fat, higher PUFA intakes.”

Recheck all references according to the journal format.

Thank you for your suggestion. We have rechecked references according to the journal format using ENDNOTE -MDPI Journals style.

Please avoid to use nouns like ‘we’ etc.

Thank you for your suggestion. In the new version of the manuscript, we have avoided using nouns like «we».

Add some tabulated data and improve the quality of paper.

Thank you for your suggestion. We created a Table and 2 Figures in order to improve the quality of paper.

Read whole article and check its grammar and language.

Thank you for your suggestions. A professional medical writer has revised the document to ensure sufficient English quality.

Comments and Suggestions for Authors from the Reviewer 3

The topic addressed is an interesting one, but for publication, in my opinion, a series of significant improvements need to be made:

- to check if the format of the journal is respected regarding the way of presentation, including bibliographic references

Thank you for your suggestion. We have rechecked references according to the journal format using ENDNOTE -MDPI Journals style.

- the analyzed database must be presented in as much detail as possible (the total number of publications and the method of selection of those that constituted sources of information), the keywords used for the search,

Thank you for your suggestion. The authors included a Table and 2 Figures in the manuscript to present more details and keeping the article easily readable.

- certain paragraphs are treated much too briefly: Vitamin D: biological actions and health benefits, Vitamin D and immune function, Vitamin D: sources;

Thank you for your suggestion. We created 2 figures in order to cover all these suggestions.

- the chapter: Vitamin D supplementation in pregnancy, nutritional benefits and/or prevention in food allergy requires additions and a tabular presentation of clinical studies that highlight the benefits of vitamin D supplementation in pregnancy

Thank you for your suggestion. We created a Table in order to improve the quality of paper.

- likewise in the case of the benefits of administering omega 3 supplements during pregnancy

Thank you for your suggestion. We created a Table in order to improve the quality of paper.

Thank you for all comments and suggestions which help us to improve our manuscript and in general our scientific activity and we hope that the new version of our manuscript will be found suitable for publication in its present form.  

Reviewer 2 Report

Vitamin D and omega-3 (fatty acid) supplementation in pregnancy for the primary prevention of food allergy in children

Comments:

In line 17 the pathogenesis of, please mention genetic factors here just like environmental factors.

How vitamin D deficiency affect T- cells and what will be its effect on child and mother health.

Add a conclusive line in the end of abstract summarizing the whole impact of your article.

In line 43 and 44 “Regarding the environment, recent studies have indicated an effect of modification of maternal exposures during pregnancy, particularly dietary habits, in the prevention of FA” please rephrase it and how dietary habits come under the environmental studies.

Please recheck and set the sentence in line 45 “Of special interest, are nutritional interventions, such as Vitamin D and omega-3 (Ω-3) supplementation during the prenatal periods”.

Add types of food allergies and their prevalence national and international level as well as add a graph representing the food allergy in different region or countries.

Add some practical examples of food allergens and their types if possible.

Rewrite this paragraph “Several studies have documented that adequate vitamin D intake during pregnancy was associated with a decreased risk of wheezing in the offspring [51-56], and other studies have investigated the association between vitamin D consumption during pregnancy and development of asthma and allergic rhinitis in the offspring [57-63]. A European Academy of Allergy and Clinical Immunology (EAACI) systematic review noted a reduction in asthma in the offspring following maternal vitamin D supplementation in pregnancy”.

Remove irrelevant material only focus on your allergens and vitamin D impact.

Explain how food allergen effects immune system?

Draw flow chart for vitamin D activation and also add thesis process site/ organ name.

Please avoid to use abbreviations and write full forms.

Add recommended dose of vitamin D for children per day.

Add chemical structure for all type of omega-3 fatty acid and draw flow chart for its mechanism.

Add full form of omega-3 fatty acid especially in headings.

Add recommended dose of omega-3 fatty acid for adults and children.

Recheck all references according to the journal format.

Please avoid to use nouns like ‘we’ etc.

Add some tabulated data and improve the quality of paper.

There is no figure or table in this article, please mention some mechanisms how Allergens are affecting the health of maternal and child.

Overall article has no sequence and other issues it can be further improved.

Read whole article and check its grammar and language.

Author Response

(The authors gave the same response as above.)

Reviewer 3 Report

The topic addressed is an interesting one, but for publication, in my opinion, a series of significant improvements need to be made:

- to check if the format of the journal is respected regarding the way of presentation, including bibliographic references

- the purpose of the article must be well specified

- the analyzed database must be presented in as much detail as possible (the total number of publications and the method of selection of those that constituted sources of information), the keywords used for the search,

- certain paragraphs are treated much too briefly: Vitamin D: biological actions and health benefits, Vitamin D and immune function, Vitamin D: sources;

- the chapter: Vitamin D supplementation in pregnancy, nutritional benefits and/or prevention in food allergy requires additions and a tabular presentation of clinical studies that highlight the benefits of vitamin D supplementation in pregnancy

- likewise in the case of the benefits of administering omega 3 supplements during pregnancy.

For publication, a major improvement is necessary, it is not impossible to achieve, but it requires serious work.

Author Response

(The authors gave the same response as above.)

Round 2

Reviewer 1 Report

The authors made a concerted effort to revise the manuscript, and it has been much improved. However, I still have 2 major concerns. First, I am unable to see the table and figures, although I see that the authors have created a legend for them. Hence, I cannot review and accept it. If the figure is pulled from one of the references, make sure that you have permission to use it.

Second, while the authors did add a few references to the type of food allergens impacted by vitamin D and omega-3 supplementation, I would strongly suggest providing more examples and details. For example, are there certain food allergies that are more impacted by vitamin D and omega-3 supplementation?

Author Response

The authors made a concerted effort to revise the manuscript, and it has been much improved. However, I still have 2 major concerns. First, I am unable to see the table and figures, although I see that the authors have created a legend for them. Hence, I cannot review and accept it. If the figure is pulled from one of the references, make sure that you have permission to use it.

We are very sorry for this inconvenience.  We have uploaded in the system a zip file containing the manuscript, cover letter, table, and 3 figures (Fig 1, Fig 2A, Fig 2B). However, to avoid any difficulties in reviewing our article, we have uploaded a PDF “manuscript plus” containing the manuscripts plus table and all figures, additionally to JPG files.

Second, while the authors did add a few references to the type of food allergens impacted by vitamin D and omega-3 supplementation, I would strongly suggest providing more examples and details. For example, are there certain food allergies that are more impacted by vitamin D and omega-3 supplementation?

Thank you for your suggestion.

The relationship between Ω3 and vitamin D supplementation and/or Vitamin D inadequacy during pregnancy and the severity of different types of FA has not been widely explored. In one of our recent studies, the consumption of multivitamins during pregnancy was shown to be correlated with a longer duration of symptoms in children with Food Protein Induced Allergic Proctocolitis (FPIAP). The RCTs, included in this review, were focused on IgE-mediated food allergy and sensitization to certain food allergens including egg, peanut.  To our knowledge, there are no other publications on vitamin D or Ω3 fatty acids in pregnancy and the severity of food allergy in offspring.  However, we added certain articles regarding relationship between cord vitamin D levels and sensitization to milk, egg, fish, and wheat.

Reviewer 2 Report

Paper can be accepted after improving its language 

Author Response

Thank you for your suggestion.

In the revised manuscript we added information regarding the literature search strategy.  Please see lines 86-96. A professional medical writer has revised the document to improve even more English quality.

Reviewer 3 Report

There is no specified database analyzed for the creation of the article. Being a review article, the keywords used to search for information and the number of articles found on different platforms should also be specified.

Author Response

(The authors gave the same response as above.)
